# Brewer's Spent Grains: Possibilities of Valorization, a Review

**Ancuța Chetrariu and Adriana Dabija ***

Faculty of Food Engineering, Stefan cel Mare University of Suceava, 720229 Suceava, Romania;
ancuta.chetrariu@fia.usv.ro
* Correspondence: adriana.dabija@fia.usv.ro; Tel.: +40-748-845-567

**Abstract:** This review was based on updated research on how to use brewer's spent grains (BSG). The use of BSG was considered both in food, as an ingredient or using value-added components derived from brewer's spent grain, or in non-food products such as pharmaceuticals, cosmetics, construction, or food packaging. BSG is a valuable source of individual components due to its high nutritional value and low cost that is worth exploiting more to reduce food waste but also to improve human health and the environment. From the bioeconomy point of view, biological resources are transformed into bioenergetically viable and economically valuable products. The pretreatment stage of BSG biomass plays an important role in the efficiency of the extraction process and the yield obtained. The pretreatments presented in this review are both conventional and modern extraction methods, such as solvent extractions or microwave-assisted extractions, ultrasonic-assisted extractions, etc.

**Keywords:** brewer's spent grain; bioeconomy; valuable compounds

## 1. Introduction

While global hunger still plays an important role affecting millions of people, there is an overproduction of food in developed countries. High production of food includes a big amount of waste. Around 90 million tons of food waste per year are produced, which causes serious environmental problems [1], but the waste reduction has become of interest to researchers, thus alternative uses can be found [2]. Food waste can be an important source of carbohydrates, proteins, lipids, and complex nutraceuticals. The bioeconomy addresses the possibilities of transforming renewable biological resources into economically and bioenergetically viable products [3]. In industries around the world, there is a need to expand the basic resources for the production of fuel, chemicals, energy, and materials, but the pretreatment of biomass is the main impediment to the development of the bioeconomy [4]. Waste reduction has a positive impact on the environment (soil, water, atmosphere) and contributes to climate change mitigation [5]. Zero hunger, responsible food consumption, and production belong to the 17 global sustainability goals of the United Nations, which represents a global action by 2030 and defines sustainable development in the three dimensions: economic, social, and environmental [3]. Product Lifecycle Assessment (LCA) is an environmental management technique adopted by the main food industry companies regarding the environmental impact of the product and the flow of materials, energy, and waste [3]. In 2018, 1.94 billion hectolitres of beer were produced globally [6]. An hL of beer results in 20 kg of wet brewers spent grain (BSG) [7,8], which means that 2018 resulted in 38.8 million tons of wet BSG.

BSG consists of layers of peel, pericarp, and seeds with residual amounts of endosperm and aleurone from barley used as raw material (Figure 1). BSG has 80% moisture, sweet taste, malt smell, and can be considered lignocellulosic material [9–11]; it is characterized by large amounts of fiber (up to

70%), including cellulose, hemicellulose, and lignin, and protein content of 25–30% [3,12,13]. BSG is a little used by-product due to its high moisture content, which makes it difficult to transport and store and makes it an unstable product conducive to microbial growth [8]. Recycling and reuse of food waste and by-products benefits both the beer industry and the environment. EUROSTAT estimates approximately 90 million tons of food is wasted in the EU/year, which means 179 kg/person [14]. EU norms encourage the extraction of valuable components (functional foods, adjuvants, pharmaceutical preparations) from food industry by-products. Until recently, food waste was of no interest, being used only for animal feed or composting, but current trends—and because it represents a cheap and valuable raw material—have brought food waste to the attention of researchers. Waste is a valuable resource that we do not yet know how to use intelligently, and converting it from problem to resource needs to be more of an interest to us. By-products and waste can become a sustainable alternative source to reduce malnutrition and hunger in developing countries [15]. There is a growing interest in finding natural resources with antioxidant activity to effectively replace synthetic antioxidants with toxic and carcinogenic effects [16]. Oxidative stress occurs as an imbalance between the number of oxidants and the antioxidant defense mechanism, and regular consumption of cereals and by-products helps prevent many chronic diseases associated with oxidative stress. Since BSG is derived from food materials, it can be incorporated into food diets (bread, snacks, muffins, pasta, etc.) [17]. The purpose of this review is to update information on BSG, health roles, and methods of extraction of valuable components.

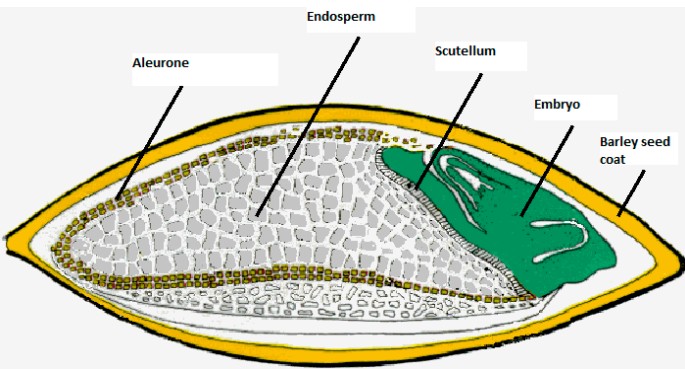

**Figure 1.** Barley structure.

## 2. BSG: A Valuable By-Product

BSG has a complex chemical composition, which varies according to variety of barley used, harvesting and malting time, and mixing time [8]. Due to the high moisture of the BSG (approximately 80%), shelf life is 7–10 days. Preservation can be done with acids (lactic acid, benzoic acid, formic acid, or acetic acid), which is contrary to the desire of consumers of the most natural food, or can be used in a mixture of benzoate-propionate-sorbate in concentrations 0.2–0.3% (v/v), which can extend its validity by 4–5 days [8]. Drying is considered the most effective method of preservation of BSG. A drying method can be used in two stages—pressing and drying—bringing moisture to 10% [9]. According to studies by Lynch et al. (2016), other preservation/drying methods that can be used are:

- drying in the oven is considered the most suitable but must be carried out at temperatures below 60 °C, with the disadvantage of a high energy consumption;
- drying by overheated steam in a thin layer brings the advantage of less consumption and improved drying efficiency;
- autoclaving at 121 °C for one hour has the disadvantage of solubilization of polysaccharides and phenolic compounds;
- drying by freezing with the disadvantage of the need for large storage spaces;

- pressing and filtration through the membrane followed by vacuum drying and drying 2 days in the air to bring moisture to 10% with the advantage that the products in which it was used no longer show microbial growth for 6 months [18].

Wet preservation can be achieved by adding salt, mixing with pomace, membrane filtration, or vacuum drying (moisture 20–30%) [13]. BSG has high proliferation post-production and for later use must be stabilized and properly stored. For long storage, a moisture of 10% is recommended.

The chemical composition of the brewer's spent grain is shown in the Table 1.

**Table 1.** Chemical composition of dry brewers spent grain *.

| Proteins | Lipids | Fibers | Carbohydrates | Lignin | Arabinoxilan (AX) | Ashes | Lyzine | Study |
|---|---|---|---|---|---|---|---|---|
| 234 mg/g | - | - | 459 mg/g | - | - | - | - | [19] |
| 24.69% | - | - | - | - | - | 4.18% | - | [16] |
| 15–28% | 5–8% | - | - | - | - | 4.5–6% | - | [13] |
| 20% | - | 50% | - | 10–28% | 40% | - | 14.30% | [8] |
| 18–35.4% | - | - | - | - | - | - | 14.30% | [14] |
| 14.2–31% | 3–13% | 59.1–74.1% | - | - | - | - | - | [20] |
| 15–26% | 3.9–10% | 70% | - | - | - | - | - | [21] |
| 19.20% | - | - | - | 22.30% | - | 4.54% | - | [22] |
| 15.4–30% | 10% | - | - | 11.9–27.8% | - | 2–5% | - | [3] |
| 31% | 9% | - | - | 16% | - | 4% | - | [18] |
| 15.3–24.6% | - | - | - | 11.9–27.8% | - | 1.2–4.6% | - | [23] |
| 22.44% | 5.3% | - | 46.52% | 19.57% | - | 3.54% | - | [24] |
| 31.81% | - | - | 3.07% | 12.72% | - | - | - | [25] |

* values are expressed of dry matter.

Fibers are the components with the greatest interest in health intake (arabinoxylans and β-glucans) but also phenolic components (hydroxycinnamic acid) [8]. Hemicelluloses consisting mainly of arabinoxylan (AX) can be present in up to 40% of the dry weight of BSG (AX is the main non-cellulosic polysaccharide present in cereals and herbs). Another important polysaccharide in BSG is cellulose, and among monosaccharides are found xylose, glucose, arabinose, and traces of rhamnose and galactose [8].

Proteins are found to be around 20% of the dry mass, and the most abundant are hordeins, glutelines, globulins, and albumins. Of the protein content, 30% are essential amino acids, the most present being lysine [8].

Lipids are found in a low percentage, of which triglycerides have a higher percentage (approximately 67% of total extract), followed by 18% fatty acids, according to del Rio et al. (2013) [26].

BSG also contains phosphorus (6000 mg/kg), calcium (3600 mg/kg), and magnesium (1900 mg/kg) [8,9], iron, copper, potassium, and manganese [12], and the vitamins present are biotin (0.1 ppm), niacin (44 ppm), choline (1800 ppm), folic acid (0.2 ppm), pantothenic acid (8.5 ppm), riboflavin (1.5 ppm), thiamine (0.7 ppm), and pyridoxine (0.7 ppm) [9,11,13]. In addition to all these components are extractives made up of waxes, gums, resins, tannins, essential oils, and other cytoplasmatic components [13].

It is of interest to separate BSG into individual components due to its high nutritional value and low cost, having applications in both food and non-food industries. When BSG was introduced into the rat diet, it had a beneficial effect on both constipation and diarrhea due to the high content of non-cellulosic polysaccharides, the small quantity of β-glucans, and the content of proteins rich in glutamine [27]. Due to high annual quantity and low cost, it is a source of interest worth exploiting [11].

## 3. Possible Uses of BSG

Due to the complex composition of BSG, there could be diverse uses, some of which are very well known while others are still developing.

### 3.1. BSG as Animal Feed

BSG is mainly used as animal feed. When used together with nitrogen sources (e.g., urea), it provides all the essential amino acids for ruminants [11,12]. BSG is suitable as feed for cattle, poultry, pigs, and fish. Used to feed the cattle, it leads to a milk production increase but decreases its lipid content [12]. BSG can be used as a source of protein in feeding *Pangasiusanodon hypophthalmus* fish, and the 50% substitution of soy flour proved most advantageous, with fish gaining weight at the end of the experiment; additionally, protein content was significantly higher, and the cost of feed decreased by 27.56%, as evidenced by Jayant et al. (2018) [28].

### 3.2. BSG in Food

Ingestion of BSG benefits human health, in such ways as: it accelerates intestinal transit and alleviates both diarrhea and constipation, decreases the incidence of gallstones, and reduces cholesterol and postprandial glucose level, all due to the protein-rich content in glutamine, non-cellulosic polysaccharides, and soluble dietary fibers [12]. BSG can be added to food to increase fiber and protein contents [29]. Due to biologically active compounds with physiological roles in the body, food fortified with BSG is considered functional food that offers health benefits and is used as an adjunct in a balanced diet [11].

The addition of functional ingredients to the bakery has been in the sights of scientists due to its ability to reduce the risk of chronic diseases beyond basic nutritional function [20]. A percentage of up to a maximum of 15% BSG can be used, as more would negatively influence sensory properties. Bread obtained from wheat flour and treated with four enzymes in which BSG (0–30%) was added had a higher shelf life as well as improved texture and volume, according to McCarthy et al. (2013) [11]. In the study on the rheological properties of the dough supplemented with BSG by Ktenoudaki et al. (2013), it was concluded that it negatively influences the texture and decreases the volume of the dough; the structure becomes dense and is negatively influenced by both the extensional biaxial viscosity and the uniaxial extensibility [30]. With the increase of the amount of BSG addition, dough strength, peak viscosity, final viscosity, and setback decrease.

When ready-to-eat snacks rich in fiber are desired, BSG [31] can be used in products with a double fiber content compared to the control samples. Studies have revealed that 10% BSG leads to baked snacks with high crispiness index (Ci), low crispiness work (Wc), and a large number of peaks during texture analysis, which means it does not adversely affect the product. BSG has a characteristic smell that contributes to the flavor of the products, but after the test acceptance, it follows that 10% BSG offers the possibility of incorporating this product into ripe snacks [32].

BSG can be added to frankfurters sausages to obtain low-fat products [11]. A maximum of 15% BSG can be added to sausages. In the case of extrusion cooking (reactive extrusion), percentages greater than 10% BSG change volume, color, texture, and structure of the products. It results in darker products with low volume, low hardness, and denser structure [8,11]. The addition of BSG to fruit juices and smoothies has a beneficial effect on increasing phenolic content and antioxidant activity [8].

In the study conducted by Spinelli et al. (2016) fish burgers prepared with various bioactive powders of BSG were obtained [33]. Polyphenols and flavonoids were extracted by supercritical $CO_2$ at 40 °C and 35 MPa. The extract was microencapsulated and dried by drying spray to prevent it from tasting unpleasant or bitter but also to protect it from the high temperatures during cooking. The 5% BSG powder (spray drying powders) gave the best ratio between the chemical properties of active powders and the sensory evaluation of samples. In this respect, antioxidant properties were also compared; the samples containing BSG had a 30% higher amount of polyphenols and 50% more flavonoids than the control sample. Attempts were made to replace feed with BSG for finishing steers fed and as a source of polyphenols, but it had no particular effects on the validity of meat [34].

BSG strengthens foods such as pasta, infant formula, meat, and meat products [14]. Cappa and Alamprese (2017) enriched fresh pasta with eggs with BSG in a quantity of 6.2 g/100 g, obtaining fiber-rich products, even if the BSG percentage was low [35]. BSG was successfully used to obtain

a fermented beverage rich in phenolic compounds, the beverage that has a shelf life regarding the bioactive components for 15 days [36].

### 3.3. BSG Used as a Substrate

Lignocellulosic materials oppose hydrolysis, which is why lingnocellulosic biomass is subject to pre-treatments that make it accessible [5], then cellulose and hemicellulose undergo hydrolysis with their transformation into sugars that can be used by microorganisms [37].

Pretreatments are expensive and have a significant impact on the environment; thus far, no pretreatment with conversion factor 100 has been found, the most used being acidic solutions (cheap and effective) using low concentrations of acids (<5%), high temperatures (120–210 °C), pressure < 10 atm, or concentrated acids (<30%), temperatures < 100 °C, and atmospheric pressure [38]. To obtain fermentation media capable of converting polysaccharides into bioactive compounds, BSG is used as the raw material in the biorefinery process, because "green chemistry" requires finding alternatives instead of thermal or chemical hydrolytic processes. One option in this respect may be pretreatment with ion fluids, followed by enzymatic hydrolysis. Delignification with cholinium-based ionic liquids was achieved with a yield of 75.89% compared to traditional delignification with imidazole with a yield of 40.18% [39].

#### 3.3.1. Substrate for the Cultivation of Microorganisms and the Production of Enzymes

BSG is mainly used for the growth of fungi for the production of enzymes (alpha-amylases, cellulases, hemicellulases), and amino acids, vitamins, and inorganic compounds are added to improve enzyme yield [8]. The microorganisms successfully grown on the medium of BSG are *Peurotus*, *Agrocybe*, *Lentinus*, and *Trichoderma,* and *Streptomyces* bacteria, according to reports of Mussatto (2009) [23]. BSG is suitable for the isolation and the maintenance of known and highly suitable strains for screening and production of new biologically active substances [17].

According to studies by Mussatto (2009), the substrate value is determined by the conversion yield of biomass [23]. An essential step in obtaining enzymes is the pretreatment of lignocelluloses, followed by enzymatic hydrolysis. The production of commercial enzymes is an expensive process [1]. Varieties of arabinoolignoxilans (AX) can be obtained with different structures through hydrolysis processes, products of interest to the food industry [1].

#### 3.3.2. A Substrate in Fermentation Processes

Production of Xylitol

Xylitol is a polyol with sweetening power similar to sucrose but with lower caloric value (2.4 versus 4.0 cal/g), thus it can also be consumed by insulin-deficient people. The downside is the higher price of sucrose that can be solved by an affordable production technology [40]. Xylitol can be produced by microbial fermentation, the BSG proving to be a more economical alternative to other lignocellulosic materials [17]. Mussatto et al. (2008) and Mussatto and Roberto (2006) optimized the process for the hydrolysis of hemicelluloses from BSG by obtaining xylitol as the final product [41,42]. The first step in obtaining xylitol is the fermentation of the BSG substrate with the help of *Candida guilliermondii* yeast [43]. From the fermentation process results $CO_2$, which is separated from the bioreactor. The broth is followed by evaporation at a temperature of 40 °C and an ethanol crystallization process to increase xylitol insolubility to 5 °C. To separate the crystallized xylitol from molasses, centrifugation is used, resulting in efficiency between 87–92%, according to studies conducted by Mussatto et al. (2013) [44]. From rice straws was obtained a yield of xylitol (percentage of the theoretical yield of 0.917 g/g) of 78.5% in agitated flasks or 57.8% in a stirred-tank bioreactor with detoxified hydrolysate [45]. From BSG was obtained 0.107 g xylitol per gram dry matter [46], and from sugar cane biogases was obtained 0.633 g/g with an efficiency of 68.97% [47]. Xylitol can be given to diabetics due to low caloric value and has beneficial properties in otitis media, osteoporosis, and lung infections [46]. Xylitol is of great

commercial importance due to its anti-cariogenic properties, preventing the formation of acids that attack tooth enamel and at the same time inducing the remineralization of enamel [47].

Production of Lactic Acid

BSG has been evaluated as the raw material for the production of lactic acid [17]. Polylactic acid (PLA) has bioplastic applications, and BSG is a precursor in obtaining it with the help of *Lactobacillus delbrueckii* from lignocellulosic material. Cellulose is subjected to a chemical treatment to make it more accessible to enzymes, enzymatic hydrolysis, which obtains a sugar solution with glucose as the main component, followed by the fermentation of hydrolysate with the help of *Lactobacillus delbrueckii* [21,48]. A similar way of producing lactic acid was also used by Mussatto et al. (2013). The process begins with enzymatic saccharification of cellulose; after, the cellulosic material is washed with water until neutralized and dried up to 10% [44]. Unconverted components are separated by vacuum filtration, and highly hydrolyzed glucose is fermented with *Lactobacillus delbrueckii*, resulting in a yield of 0.96 g lactic acid per gram glucose. In another study, Mussatto et al. (2008) obtained a lactic acid yield of 0.98 g/g glucose, similar to the theoretical maximum value of 1 g/g glucose [41]. Other microorganisms used to obtain lactic acid are *Lactobacillus pentosus* and *Lactobacillus rhamnosus* [23].

Obtaining lactic acid offers great opportunities in obtaining biodegradable polymers as plant growth regulators or in obtaining green chemicals/solvents [21]. Liang and Wan (2015) used BSG substrates to obtain carboxylic acids with mixed-culture fermentation, a study in which lactic acid was the dominant one in both acidic pH (9.2 g/L) and alkaline pH (6.7 g/L) [49].

Ethanol Production

Lignocellulosic hydrolysates can be used as a fermentation medium for obtaining ethanol, xylitol, and other products, which must be subjected to detoxification before fermentation to improve the efficiency of the process [37]. Applied treatments can be treatments with acids, microwave digestion, ultrasound, or enzymatic hydrolysis to make glucose extraction from cellulose easier [12]. Celluloses break down into glucose, and hemicelluloses break down into xylose, arabinosis, manosis, glucose, acetic acid, and galactose [37], which can be converted with microorganisms into ethanol. Two methods are used: separate fermentation hydrolysis (SHF) and hydrolysis and simultaneous fermentation (SSF) [1]. Mussatto and Roberto (2006) used an acid hydrolysis process resulting in a maximum concentration of 13.21 g/dm$^3$ xylose, 8.21 g/dm$^3$ arabinosis, and 0.31 g/dm$^3$ glucose under the conditions of a maximum sugar yield of 19.6 g xylose/100 g dry matter and 8.3 g arabinosis/100 g dry matter [42]. According to studies reported by Mussatto (2014), the production of combustible ethanol takes place in five steps: BSG pretreatment (with acid heat solutions), hydrolysis of BSG with enzymes to convert starch and cellulose into simple sugars, fermentation of sugars with ethanol result (with the help of *Zymomonas mobilis*, *Saccharomyces cerevisiae*, *Escherichia coli*, *Bacillus subtilis*, or *Pichia pastoris*), followed by ethanol distillation and its dehydration to remove its water [12]. Dragone et al. (2007) achieved maximum productivity of 2.09 g.L$^{-1}$.h$^{-1}$ ethanol for wort with 19.6 °P by continuous fermentation versus a maximum of 0.47 g.L$^{-1}$.h$^{-1}$ ethanol for wort with 20 °P by discontinuous fermentation [50].

Lignocellulosic Yeast Carrier (LCYC)

The continuous fermentation of beer is based on a population of high-density yeasts, most commonly obtained by attachment to solid substrates with the formation of biofilm. The important properties of the solid substrates are increased adhesion capacity, physical endurance, high availability, and low-cost of production [51]. Continuous fermentation systems represent an innovative process for obtaining products of uniform quality, and beer production is faster. Continuous fermentation systems with immobilized cells reduce the time of primary fermentation to 1 day, but the cost of immobilization support is high and is the main impediment in industrial application [50,52]. BSG is a good source for lignocellulosic yeast (LCYC) [17], because it is a residue of the beer industry, thus it can be easily integrated into the continuous fermentation of beer. BSG is a low-cost material with good stability and

high retention capacity of yeast cells; it can be easily prepared without modifying the chemical content, then sterilized and regenerated by washing in a caustic solution. No investments are necessary, it being a byproduct of the beer industry [52,53], and it represents an alternative worth exploring in support of immobilized cells [52]. Studies say that protein and fat extraction is recommended both for obtaining valuable fractions of BSG and for improving the LCYC production yield [51].

### 3.3.3. Prebiotics

Xylan is the main component of hemicellulose present in the plant cell walls. Xilooligosaccharides (XOS) are mainly produced by hydrolysis of xylan, and the prebiotic effects have been demonstrated. According to Amorim et al. (2019), XOS shows stability to heat and pH, and its use in food gives acceptable organoleptic properties [54]. It has beneficial effects on health, namely in the prevention of diabetes, neurotoxicity, inflammation of the colon, detoxification, weight loss, treatment of constipation, and microencapsulation. The BSG lignocellulosic biomass is composed mainly of cellulose, hemicellulose, and lignin. Cellulose is composed of a chain of glucose molecules with hydrogen bonds between different layers of polysaccharides, giving a crystalline conformation. Hemicelluloses, consisting mainly of xylan, are the target component for XOS production. Lignin supports the integrity and the rigidity of cell walls [9], and it is composed of phenolic components (*p*-coumaryl alcohol, coniferyl alcohol, and sinapyl alcohol) [54]. BSG contains hemicelluloses in a percentage of 16.5%, composed mainly of xylan (10.3%), arabinan (5.1%), and a small group of acetyl groups (1.1%) [55]. Using a substrate of BSG, Amorim et al. (2019) obtained, with commercial xylanoses with *Trichoderma longibrachiatum,* 444.3 mg XOS/g xylan in 12 h, and with *Trichoderma reesei* by direct fermentation, 326.2 mg XOS/g xylan in 72 h, without prior treatment [54].

A novelty for the food industry is food containing symbiotic products, both probiotic bifidobacteria as well as prebiotic oligosaccharides, products enriched both in terms of physico-chemical properties and in terms of health benefits. Non-digestible oligosaccharides have potential as food ingredients to improve the quality of many foods in terms of flavor and physico-chemical characteristics [56].

### 3.4. Obtaining Building Materials (Bricks)

The high amount of fibrous material combined with the reduced amount of ash makes BSG suitable for obtaining bricks [13,17]. The sawdust is commonly used in obtaining bricks and can be replaced with BSG due to the increase in the porosity of the bricks without changing their color or quality [13].

### 3.5. Adsorbent

BSG has been tested as an adsorbent for the removal of volatile compounds from waste gases but also for heavy metals from aqueous solutions (cadmium, lead, or chromium adsorption at adsorption capacities of 17.3 mg/g, 35.5 mg/g, and 18.94 mg/g) [12,17]. Pyrolysed BSG can absorb volatile organic compounds similar to coal from coconut shells [12].

BSG has also been successfully used as an adsorbent of orange acid dye 7 in wastewater, with an adsorption capacity of 30.5 mg of orange acid 7 per gram BSG at 30 °C [12].

Activated carbon can be obtained using BSG as raw material, a material used to purify water and gas. It is obtained by an alkaline treatment applied to BSG for the recovery of lignin, followed by precipitation with sulphuric acid and impregnation/activation of recovered lignin with 3 g per gram phosphoric acid at 600 °C and has the ability to adsorption of phenolic compounds and metal ions in liquid media [12].

### 3.6. Source of Phenolic Compounds

Phenolic compounds in plants are made up of several components: phenolic acids, phenolic alcohols, flavonoids, tannins, stilbene, and lignans [57,58]. BSG is a generous source of hydroxycinnamic acids (HCA) [9], which accumulate in cell walls are considered the most important source of antioxidants

in cereals, both in free form and the bound form [17]. Phenolic compounds are considered natural antioxidants associated with some chronic diseases, such as cardiovascular disease, neurodegenerative diabetes, and cancer [9].

Phenolic acids, including caffeic acid, vanillic acid, quercetin, and epigallocatechin-3-gallate, inhibit cyclooxygenase isoforma, reducing the risk of cancer according to the method evaluated by DNA fragmentation and the Hoechst staining test [11]. Phenolic acids are grouped into hydroxycinnamic acids (HCA) and hydroxybenzoic acids (HBA), the latter in low amounts in BSG. Ferulic acid (FA) and p-cumaric acid (*p*-CA) are the most abundant HCAs in BSG, with values between 35–490 mg/100 g dry matter for FA and 6.7–180 mg/100 g dry matter for *p*-CA [17]. Ferulic acid is the target product of BSG; it is an antioxidant found in cell walls with a key role in the development and the protection of the plant. Ferulic acid has similar properties in the human body, protecting the skin and keeping it young, with effects similar to vitamin C [1]. It is a stable and gentle compound with the skin that reduces the negative effects of free radicals, considerably slowing the aging of the skin. It has also been approved as a preservative in many countries to prevent food oxidation [9,11]. For these reasons, extraction and recovery of these compounds is of niche interest for research for both the food industry and the pharmaceutical/cosmetic industry [17].

The important steps in phenolic compounds extraction include pretreatment, extraction, isolation, and purification. There is an upward trend of "green chemistry" or "green extraction" that implies the use of solvents and substances obtained from renewable materials but which could keep the quality of the gained extraction [57]. The extraction methods and the used treatments are microwave-assisted extraction (MAE) and saponification with NaOH 1–4 M (alkaline hydrolysis is the most used). Contents of 27.31 ± 0.69 μg/mL ferulic acid and 0.732 ± 0.020 mg galic acid equivalent (GAE)/mL phenolic acid were obtained at a concentration of 1 M. Other methods imply acid hydrolysis, enzymatic hydrolysis, ultrasound extraction, liquid–liquid extraction or solid–liquid extraction, water bath extraction with solvents, supercritical $CO_2$ extraction, and cosolvent ethanol (extraction temperature 40 °C, 240 min, 35 MPa $CO_2$ pressure, and 60% ethanol [9,17].

Enzymatic hydrolysis proved to be less efficient and with a higher cost of the analyses due to the high cost of pure enzymes (cellulase, $\alpha$-amylase, pectinase) [17]. Microwave-assisted extraction (MAE) is an efficient, promising, and fast technique. Microwave hydrodiffusion and gravity (MHG) is a new bioactive compounds plant extraction technique that mixes the microwave heating with the gravity, the whole process deploying at atmospheric pressure without the water implication or another solvent [57]. Solid-state fermentation (SSF) is when the microbial growth and formation of products take place almost in the absence of water; the substrate contains moisture to allow growth and metabolism of microorganisms. Submerged fermentation (SmF) assumes that microorganisms are grown in a liquid medium containing the necessary nutrients [59]. Studies have shown that SSF has a higher yield than SmF in the production of bioactive compounds in addition to lower costs and secondary compounds that occur in a shorter time than SmF without the need for aseptic conditions. Very important roles in the case of these fermentations lie with humidity and water activity, which show us the water available for microbial growth, which affects the development of biomass, metabolic reactions, and mass transfer processes [59]. According to studies carried out by Moreira et al. (2013), up to 20 mg GAE/g BSG can be extracted [51]. The most important factor in the extraction of phenolic compounds is the solvent used. Meneses et al. (2013) reported a total phenolic compound content of 9.90 ± 0.41 mg GAE/g dry matter, using 60% acetone (*v/v*) and 2.14 mg GAE/g BSG when ethyl acetate was used, with an increase in extraction yield of 4.6 times [16]. When acetone was used as a solvent, the flavonoid content was between 0.51–2.12 mg QE/g BSG. Guido and Moreira (2017) reported results of 1.26 ± 0.10 and 4.53 ± 0.16 mg GAE/g dry matter for pale BSG and dark BSG (obtained by drying at very high temperatures), using acidified methanol as a solvent (HCl/methanol/water 1:80:10 *v/v/v*) [17], while Moreira et al. (2013) reported contents of 19.5 ± 0.6 mg GAE/g dry matter for BSG light and 16.2 ± 0.6 mg GAE/g dry matter for BSG dark [51]. The type of malt used and the malting process (burning regime and roasting temperatures) influence the content of phenolic compounds, BSG light

having a higher content than BSG dark [51]. According to studies by Vellingri et al. (2014), the total content of polyphenols in BSG was 167.07 mg GAE/l following a first extraction with 80% ethanol and 66.75 mg GAE/l following a second extraction with the same solvent [60]. When the extract is used in food, the solvents used are very important and are regulated from the perspective of food safety (according to good manufacturing practices are the mixture acetones:water or ethanol:water). Methanol is a very effective extraction solvent, but its toxic nature limits the use of the extract in the food industry or in the pharmaceutical industry [16]. An alternative and ecological method to conventional extractions with organic solvents is the method of supercritical $CO_2$ extract (SCE-$CO_2$), which uses $CO_2$ in a supercritical condition when both temperature and pressure are equal to or exceed the critical point of 31 °C and 73 atm, providing the ideal conditions for extracting compounds with a high degree of recovery in a short time [61]. BSG has a lower content of phenolic compounds than berries, which have a content between 28–51 mg GAE/g, but its phenolic content is much higher than many vegetables (e.g., onions 2.5 mg GAE/g, potato peel 4.3 mg GAE/g, tomatoes 2.0 mg GAE/g), making it an interesting by-product to evaluate [16]. An interesting application is an increase in the content of bioactive phenolic compounds in food, which have a big impact lately due to the desire of consumers to improve their health through food [59].

### 3.7. Biogas Production

The whole world is going through an energy crisis, a crisis that can be reduced by producing energy from the BSG due to its high availability and low cost [12]. Biogas refers to a mixture of gases produced by anaerobic decomposition of organic matter through a complex process that occurs naturally in an oxygen-free environment and that is considered an effective method of converting biomass into methane [62]. Biogas is composed of methane (40–75%), water (0–10%), carbon dioxide (25–55%), hydrogen sulfide (1–3%), ammonia 0–1%, nitrogen 0–5%, oxygen 0–1, and hydrogen 0–1%. Biogas has a thermal value of approximately 22 MJ/$m^3$ [63], is considered a clean, $CO_2$-free fuel, and can be used in fuel cells for electricity generation. Although it is a recyclable, efficient, and clean product, 96% of its production is made with fossil combustibles, resulting in environmental pollution and energy crises due to high energy consumption [64]. Obtaining biogas from BSG involves largely two steps: a hydrolytic stage that allows a complete degradation of the material, a very important step to obtain high yields, and a methanogenic stage, where, with the help of macromolecule microorganisms, they convert into volatile fatty acids, acetates, butyrate, propionate, and methane [12]. Pretreatments play an important role in the degradation of the crystalline structure of cellulose molecules and decrease the degree of polymerization, easing enzymatic hydrolysis into simple sugars. Alkaline pretreatment provides a pH-friendly environment for further fermentation that is more effective [64]. After 15 days of digestion in a batch of anaerobic fermentation, the yield of biogas was 3476 $cm^3$ per 100 g BSG, according to Mussatto studies (2014) [12]. A decisive factor in hydrolysis and the use of lignocellulosic biomass in the production of biofuels is crystallinity (the crystalline index indicates the amount of crystalline cellulose present in biomass). Zhang and Zang (2017) used BSG pretreated with calcined red mud (resulting from bauxite), which reduced the BSG's crystalline index, with positive effects on BSG solubility and biohydrogen production [64].

### 3.8. Food/Composite Packaging

Incorporation of chitosan into BSG proteins can give rise to microfilm with antimicrobial and antioxidant properties, according to food packaging [14]. The development of biological and biodegradable materials is influenced by the ability of BSG proteins to interact between polypeptide chains. It is necessary to add plasticizers to protein-based films, which play a role in reducing the fragility of film and giving it certain plastic properties, which helps to increase flexibility and film handling, for example, sorbitol, polyethylene glycol, and glycerol. Film formation is influenced by protein–protein interactions, which are more intense at pH close to the isoelectric point. According to studies conducted by Proaño et al. (2020), the films formed at pH = 2 were the most homogeneous

and could easily detach from the support [65]. Films obtained with polyethylene glycol (PEG) as plasticizer show homogeneity and are fine. Water solubility increases with increasing the amount of PE. The films also show good properties of water barrier, color is influenced by protein concentration, and opacity increases with the increase of PEG. Opacity is very important when packing fatty foods, as the oxidative degradation given by light is alleviated. The obtained films can be used as a barrier against UV, as they do not present transmittance between 200 and 400 nm. The films obtained present antioxidant activity to be taken into account in the case of active packaging of food. Not many studies have been conducted on this segment of obtaining biodegradable films from BSG, thus it is a subject to consider.

Cellulose is used for food packaging due to its fine network, biodegradability, and high water resistance. It can be added as a source of fiber or as a thickening and stabilizing agent in functional foods and drinks. Cellulose can also be used to produce nanocomposite materials [4]. Natural cellulose-rich fibers, including BSG, can be used as an alternative to produce organic polyurethane, which has proven to be the most promising filler. Different BSG polyurethane rations and ground tire rubber slots have been studied by Formela et al. (2017), with improvements in physicomechanical properties (apparent density increases by 37%, compression resistance by 50%) and thermal stability [66].

### 3.9. Proteins, Protein Hydrolysates, Bioactive Peptides

Because of the upward trend of vegetarianism and veganism, it is necessary to find sources of protein and natural protein derivatives from plants. There are several possibilities of using proteins derived from plants due to the functional effects on the human body (related to physiological and nutritional properties) and technical positive properties (related to physicochemical properties: appearance, texture, stability) as well as for nutrients for fortified foods and dietary supplements, technical-functional ingredients in terms of emulsification and gelification properties, or as materials for the development of biopolymers [67]. The application of proteins from BSG include antioxidant and antimicrobial packaging materials production, edible films (obtained from proteins and polysaccharides/ chitosan), complexing proteins, fermented beverages, cookies, enzymes, and edible proteins. Proteins and protein hydrolysates derived from BSG, as resulted from the study of Ikram et al. (2017) [9], present immunomodulatory effects that can be useful to anti-inflammatory diseases control, hypertension, and diabetes treatment [8]. The most abundant protein in BSG is hordein, with a content of 43% of the total content of proteins, followed by glutelins with 21.5% [9].

Amino acids play an important role in human health, and BSG is a good source of these components. Essential amino acids from BSG are methionine, phenylalanine, tryptophan, histidine, and lysine, and the non-essential are serine, alanine, glycine, and proline [9]. A study conducted by Wen et al. (2019) revealed the fact that BSG has a similar composition in amino acids as in germinated barley—rich in glutamine/glutamic acid, vanillin, and leucine but with a smaller content of cysteine and methionine [68].

According to the studies of Pojicet et al. (2018), protein products can be classified in:

- protein containing flour up to 65% protein;
- protein hydrolysates containing concentrates between 65–90% protein;
- protein isolates containing more than 90% protein [67].

For an increased extraction yield, pretreatment BSG is applied (with diluted alkaline, enzymatic, or hydrothermal acids, or combinations thereof). Pretreatment with diluted acids can extract 90% of the total protein; it is not a selective method, extracting carbohydrates and lignin along with proteins. Hydrothermal pretreatment offers this selectivity, although the extraction yield is 64–66%, but it is a more environmentally friendly method, thus it is carried out at low temperatures and does not require the addition of chemicals [69]. The extraction methods used are largely synthesized in the following categories:

- dry extraction techniques: fractions with high impurity and agglomerated particles. This category includes two-step electrostatic separation, which involves loading particles and separating them in an electric field [67];
- wet extraction techniques: acidic extractions are less effective because they do not degrade the cell wall, resulting in less protein in the extraction environment. Alkaline extractions are more effective but at high alkaline concentrations after the Maillard reaction, which affects the nutritional properties of proteins. Extracts with organic solvents are also used [68]. Combinations of water with enzymes, water in subcritical conditions, or protein extraction through reverse smalls attract more and more of the attention of researchers [67]. Alkalis are used as extraction solvents at high temperatures, followed by precipitation with alcohol or isoelectric precipitation, treatment with NaOH/KOH (0.1 M, 0.5 M, 4 M) for 24 h at room temperature, acidification with citric acid up to pH = 3, then precipitation with ethanol, an enzymatic hydrolysis-effective technique for extractions [9];
- other extraction methods are microwave-assisted extraction (MAE), ultrasonic-assisted extraction (EAU), electrically pulsed energy extraction, or extraction using high hydrostatic pressure [67].

Treatment of BSG with sodium dodecyl sulfate (SDS) and disodium phosphate at different temperatures, followed by the addition of ethanol and refrigeration of extracts, resulted in a yield of 49% protein [70]. Bioactive peptides (BAP) are specific amino acid sequences that bring health benefits of interest to the pharmaceutical industry in terms of the discovery of new nutraceuticals or the food industry as ingredients for functional foods [15]. Peptides derived from BSG have potential against cardiovascular diseases and type 2 diabetes [71]. The specific bioactivity of food peptides is given by the length of the amino acid chain and their hydrophobicity and molecular weight. There is a growing interest in the use of bioactive peptides derived from food proteins against chronic diseases and for maintaining health status [72], inhibitory activity on angiotensin conversion enzyme (ACE), and dipeptide peptidase IV (DPP IV), which is being studied in vitro by Cermeno et al. (2019) [71]. BAPs are encrypted in the primary structure of proteins in the form of inactive amino acids (inactive when part of the source protein) but are activated by fermentation and food processing with the help of enzymes or in the digestive tract after human consumption [72,73]. According to the BIOPEP-UWM Database of Bioactive Peptides [74], the biological functions of bioactive peptides are ACE inhibitors (angiotensin conversion enzyme I), influence blood pressure regulation, antioxidant activity, antimicrobial activity, opioid activity, and antithrombotic and immunomodulatory activities. Strong bioactivity and high yield are challenges in BAP research [72]. The introduction of bioactive peptides into functional food brings the benefits of the amino acids contained. The techniques used for the synthesis of bioactive peptides are chemical synthesis, enzyme synthesis (as the main technique, microbial fermentation of proteins or protein isolates) [75], and synthesis by recombinant DNA technology [15]. Protein hydrolysis is not only a way to improve nutritional values but also to release bioactive peptides [75]. Obtaining pure peptides involves a laborious process of splitting and isolating individual peptides by chromatographic methods and membrane technologies [75].

Studies in which hydrolyzed proteins in cereals have increased the shelf life of meat have been conducted as the incorporation of antioxidant peptide fractions decreased lipid oxidation from 19% to 15% after one week [75]. Protein hydrolysates can be used in the food industry as texture improvement agents and as food additives or in the pharmaceutical industry. Hydrolysis in BSG has had good rheological results; protein changes by enzymatic or chemical means bring improvements to certain functional properties (water/oil retention capacity, emulsion properties and foam expansion, turbidity) [76]. Hydrolyzed protein isolates with a number of enzymes (Alcalase, Corolase PP, Flavourzyme, and Promod 144 MG) had improvements in heat stability at pH = 6.0, even after 300 min of temperature maintenance at pH between 2.0 and 12.0, which increased solubility, emulsification, and foaming capacity.

Nitrogen solubility decreased when pH approaches isoelectric point (pH = 3.8), but the highest solubility was achieved at pH between 6.0 and 12.0. Compared to non-hydrolyzed soy protein, which is

more stable at pH = 9.0 than at pH = 3.8, the protein hydrolysates in BSG showed no coagulation at pH = 8.0, but HCT (heat coagulation time) decreased at pH below 6.0 [77]. Protein isolates and protein hydrolysates associated with them can be used as techno-functional ingredients but carefully, as the pH at which they are used influences these properties according to the above-mentioned study.

### 3.10. Source of Fiber

BSG is a rich source of dietary fiber, especially viscous fiber, with a significant contribution in speeding up intestinal transit and improving the symptoms of ulcerative colitis [9]. Dietary fibers are classified according to solubility: soluble dietary fibers include β-glucans pectic polysaccharides, arabinogalactans, and xyloglucans, and insoluble dietary fibers include lignin, cellulose, and galactomannans [9].

Lignin can be partially degraded by intestinal microbiota, and the resulting compounds can be metabolized, according to studies by Lynch et al. (2016) [8]. Berglund et al. (2016) conducted studies to obtain cellulosic nanofibers using a method of bleaching and mechanical separation but obtained low yields (22%) compared to carrot residues from juice processing (32%) [78]. Energy consumptions were high at 21 kWh/kg and 0.9 kWh/kg, respectively.

The method of separating proteins and fibers proposed by He et al. (2019) involves grinding with a disk mill, then mixing the BSG with deionized water [79]. Then, one adds the reagent (NaOH, sodium bisulfite, or alcalase) to the suspension on the water bath for 4 h at 60 °C, then transfers the samples to a sieve shaker where it is shaken for 15 min to separate the small solubilized proteins from the large insoluble fibers. During the sifting, the samples are washed with deionized water, the proteins pass through the sieve, and the fibers remain on the sieve.

### 3.11. Polymers

Food waste may be a good source for the production of polyhydroxyalcanates (PHA) and poly-3-hydroxybutyrate (PHB) due to abundance and low cost. PHA is a material similar to plastics and can replace plastics derived from oil. The main disadvantage of producing PHA is the high operational cost of production, reduced by the use of raw materials resulting from food waste, such as BSG [1]. The production of volatile fatty acids without pretreatment by anaerobic digestion helps to obtain bioplastic materials (e.g., PHA) by using BSG as a substrate [80].

### 3.12. Other Application

BSG can also be used for obtaining paper-serves-cards, production of coal (inferior in terms of burning properties), production of resins, production of an antifoaming agent in beer [12,13], and production of bacteriocin using *Lactococcus lactis* and *Enterococcus mundtii* with antimicrobial activity against *Lysteria monocytogenes* as a bioindicator. Lignocellulosic material is subjected to treatment, usually hydrolysis, in order to solubilize in the constituent monomers. The sugars formed are subject to microbial fermentation as precursors of added-value compounds (e.g., bacteriocins) or enzymes [25]. Ndayishimiye et al. (2020) conducted a study on the encapsulation of oils recovered from BSG by an eco-friendly technique of saturated gas solutions ($CO_2$ supercritical), aiming to obtain a product with improved physical properties and with oxidative stability [81].

The growing demand for products obtained with stable ingredients obtained from food by-products stimulates the finding of innovative alternatives to obtain those [81]. The concept of biorefinery says that all components of raw material are converted into products of commercial importance (e.g., biofuel, enzymes, oils, nutraceuticals) [82] and is adopted in many sectors for integrated food production. Mussatto and coworkers (2013) developed an integrated system for the production of lactic acid, xylitol, activated carbon, and phenolic acids [44]. Bio-based economy is an economy based on the use of feedstock biomass for food, energy, chemicals, and other materials. The use of biomass as a raw material for the production of the above mentioned brings social benefits and economic potential and ensures a reduction in carbon emissions [5].

Pyrolyzed BSG is used to obtain biochar, which is a porous material with stable physical and chemical properties and a high content of C. It is used to increase the amount of C in the soil and to reduce nutrient leakage; it is also used for soil amendment and as a nutrient supplier for plant growth and improving soil characteristics. Biochar has a large amount of ash, stable aromatic C structures, low bulk density, and moderate cation exchange capacity, which makes it suitable for this use [83]. The submerged fermentation of BSG using *Aspergillus niger* and *Saccharomyces cerevisiae* can obtain citric acid in amounts of 0.512% and 0.312%, respectively [84–86].

## 4. Conclusions

The use of food waste brings benefits to both reducing environmental pollution and to industry. Turning by-products into value-added components reduces food production costs and quantifies their nutritional value. It is necessary to find low-cost sources of valuable compounds of plant origin given the upward trend of vegetarian and vegan diets. The researchers seek new alternatives for food fortification, and the conversion of vegetable by-products in higher value products for obtaining functional compounds has their attention, with researchers showing more and more interest.

**Author Contributions:** A.C. and A.D. contributed equally to the collection of data and preparation of the paper. All authors have read and agreed to the published version of the manuscript.

**Funding:** This research received no external funding.

**Conflicts of Interest:** The authors declare no conflict of interest.

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
