# Peer review of "Brewer’s Spent Grains: Possibilities of Valorization, a Review"

_applsci, doi:10.3390/app10165619_

Round 1

Reviewer 1 Report

In the manuscript “Brewer’s spent grains: possibilities of valorization. The Review, ”the authors gather updated information regarding Brewer’s spent grains, their chemical composition, extracting methods, valuable components, nutritional properties, and their applications in both the food and non-food industries.

The authors make a careful review of the subject, having cited 86 references.

I think the paper would have a lot to gain by including some figures. For example, line 43, the authors describe the constitution of Brewer’s spent grains, do not indicate which species they are describing they are describing (I think it’s wheat, the world’s most important cereal to produce beer). If they put an illustration their identification would be immediate.

Line 88, remove “as it results from the studies of several authors.”

Line 120, remove “Due to the high annual quantity and low cost, BSG is a source of interest worth exploiting. Bioeconomically BSG can be used in food and feed, materials and chemicals, bioenergy and biofuels.”, because it has been said before.

Line 130, the species name (Pangasiusanodon hypophthalmus) must be written in italics.

Line 387, remove “The analysis of total phenolic compounds is done mainly with the Folin-Ciocalteau method, the simple, convenient method that requires equipment that is at the disposal of laboratories, but also by Differential Pulse Voltametry (DPV) [57]. The majority phenolic compounds are determined by HPLC or thin-layer chromatography [57]. Antioxidant activity is determined by the DPPH method (2,2-diphenyl-1-picrilylhydrazil), by the absorption capacity of the oxygen radical (THE FRAP/ORAC method) or by the TEAC-trolox equivalent of antioxidant capacity.”, because it has no relevance.

A revision of the text is important to avoid some mistakes as in line10 “based on updating research” change to “based on updated research”, line27 “which cause serious” change to “which causes serious”, line 32 “industries around the world there is a need to” change to "industries around the world, there is a need to "; line 54 "waste is valuable" change to " waste is a valuable ", line 142 " food that offer health" change to "food that offers health", among others.

In my opinion, with these major revisions, the manuscript will be ready for publishing.

Author Response

Reviewer: I think the paper would have a lot to gain by including some figures. For example, line 43, the authors describe the constitution of Brewer’s spent grains, do not indicate which species they are describing they are describing (I think it’s wheat, the world’s most important cereal to produce beer). If they put an illustration their identification would be immediate.
Response: First of all we would like to thank the referee for the close reading and for all the given comments suitable for improving the manuscript. Was corrected according to referee suggestions and we have included a FIGURE that exemplifies the BSG components (Figure 1).

Reviewer: Line 88, remove “as it results from the studies of several authors.”

Response: We would like to thank to the referee for his remarks. We made the changes according to referee suggestions. Reviewer: Line 120, remove “Due to the high annual quantity and low cost, BSG is a source of interest worth exploiting. Bioeconomically BSG can be used in food and feed, materials and chemicals, bioenergy and biofuels.”, because it has been said before. Response: We would like to thank to the referee for his remarks. Was corrected according to referee suggestions.

Reviewer: Line 130, the species name (Pangasiusanodon hypophthalmus) must be written in italics.

Response: We changed according to referee suggestions.

Reviewer: Line 387, remove “The analysis of total phenolic compounds is done mainly with the Folin-Ciocalteau method, the simple, convenient method that requires equipment that is at the disposal of laboratories, but also by Differential Pulse Voltametry (DPV) [57]. The majority phenolic compounds are determined by HPLC or thin-layer chromatography [57]. Antioxidant activity is determined by the DPPH method (2,2-diphenyl-1-picrilylhydrazil), by the absorption capacity of the oxygen radical (THE FRAP/ORAC method) or by the TEAC-trolox equivalent of antioxidant capacity.”, because it has no relevance.

Response: Was corrected according to referee suggestions.

Reviewer: A revision of the text is important to avoid some mistakes as in line10 “based on updating research” change to “based on updated research”, line27 “which cause serious” change to “which causes serious”, line 32 “industries around the world there is a need to” change to "industries around the world, there is a need to "; line 54 "waste is valuable" change to " waste is a valuable ", line 142 " food that offer health" change to "food that offers health", among others.

Response: We would like to thank to the referee for his remarks. We made the changes according to referee suggestions.  We improved English language.

Reviewer 2 Report

The authors wrote a review about the valorization of brewer's spent grains. 

The overview is interesting. 

My remarks: 

  1.  Line 130: italicize the latin name. 
  2.  Line 278: define what does LCYC stand for. It is in the title and can make the reader confused.

Overall impression is that this is a thoroughly studied and written review. The authors spent a lot of time reading different manuscripts, from food industry utilization of BSG to brick-making (this was an interestin discovery from my part). I recommend acceptance after an English language check up. 

Author Response

Reviewer: Line 130: italicize the latin name. 
Response: First of all we would like to thank the referee for the close reading and for all the given comments suitable for improving the manuscript. Was corrected according to referee suggestions. Reviewer: Line 278: define what does LCYC stand for. It is in the title and can make the reader confused. Response: We would like to thank to the referee for his remarks. We make the changes according to referee suggestions. We improved English language.
